biochemistry/ecology/microbiology

greenhouse camel cricket, hide beetle, *Cedecea lapagei*, lignocellulose, secretome

# Public questions spur the discovery of new bacterial species associated with lignin bioconversion of industrial waste

Stephanie L. Mathews[1], Mary Jane Epps[2],
R. Kevin Blackburn[3], Michael B. Goshe[3],
Amy M. Grunden[4] and Robert R. Dunn[5,6]

[1]Department of Biological Sciences, Campbell University, Buies Creek, NC 27506, USA
[2]Department of Biology, Mary Baldwin University, Staunton, VA 24401, USA
[3]Department of Molecular and Structural Biochemistry, [4]Department of Plant and Microbial Biology, and [5]Department of Applied Ecology, North Carolina State University, Raleigh, NC 27695, USA
[6]Center for Macroecology, Evolution and Climate, University of Copenhagen, Copenhagen, 2100 Denmark

SLM, 0000-0001-5909-2024; RKB, 0000-0001-8623-3902; RRD, 0000-0002-6030-4837

**Author for correspondence:**
Robert R. Dunn
e-mail: rrdunn@ncsu.edu

A citizen science project found that the greenhouse camel cricket (*Diestrammena asynamora*) is common in North American homes. Public response was to wonder 'what good are they anyway?' and ecology and evolution guided the search for potential benefit. We predicted that camel crickets and similar household species would likely host bacteria with the ability to degrade recalcitrant carbon compounds. Lignocellulose is particularly relevant as it is difficult to degrade yet is an important feedstock for pulp and paper, chemical and biofuel industries. We screened gut bacteria of greenhouse camel crickets and another household insect, the hide beetle (*Dermestes maculatus*) for the ability to grow on and degrade lignocellulose components as well as the lignocellulose-derived industrial waste product black liquor. From three greenhouse camel crickets and three hide beetles, 14 bacterial strains were identified that were capable of growth on lignocellulosic components, including lignin. *Cedecea lapagei* was selected for further study due to growth on most lignocellulose components. The *C. lapagei* secretome was identified using LC/MS/MS analysis. This work demonstrates a novel source of lignocellulose-degrading bacteria and introduces an effective workflow to identify bacterial enzymes for transforming industrial waste into value-added products. More generally, our research suggests the value of ecologically guided discovery of novel organisms.

# 1. Background

Humans continue to produce industrial compounds that are relatively hard to degrade. In some cases, these compounds, including plastics, are intentionally hard to degrade so as to increase the longevity of their usefulness. In other cases, the longevity of these products is unintentional, as is the case with many industrial byproducts. The total quantity of industrial waste recalcitrant to ordinary decomposition is immense, accelerating in its rate of accumulation and problematic for local and regional economies, wild ecosystems and human health [1,2]. While we need to reduce the production and accumulation of these recalcitrant compounds in the future, there is also a pressing need to find ways to degrade those that have already accumulated in landfills and natural ecosystems. One promising strategy for dealing with this accumulated waste is to find microbes and their genes that are better able to degrade such compounds [3]. Here we describe research in which we use insights from ecology, evolutionary biology and public engagement to discover bacteria able to degrade one particularly problematic industrial waste product [3], black liquor and its most abundant component, lignin.

Black liquor is a waste product from the paper pulp industry that is commonly produced during the chemical breakdown of pulped wood. It consists primarily of lignin and hemicellulose (components of lignocellulose that make up plant cell walls) suspended in a highly alkaline solution [3]. In countries in which environmental legislation is lax, black liquor is sometimes removed from paper production facilities as waste runoff into waterways, resulting in the death of fish [4–6]. In the USA (where regulations are strict, but control is lax) black liquor is more often burned in boilers to recover some energy in the form of steam energy but with the loss of the fixed carbon that could be converted to commodity chemicals. Emissions from recovery boilers contain volatile organic carbon and are associated with respiratory illnesses and premature mortality [7]. These emissions are often odorous due to the presence of sulfur compounds, which impacts the quality of life of those living near pulp and paper mills [8]. In the long term, the ability to degrade black liquor microbiologically would have the potential to simultaneously eliminate waste and generate valorized chemicals such as alcohols and organic acids. The challenge, however, is to identify bacteria able to accomplish this task. Even though lignocellulose is the most abundant biopolymer on Earth [9], to date only a few bacterial species have been identified that are able to fully degrade this compound (including its lignin component), and only one of these is known to do so under alkaline conditions [3,10]. Microbial lignin degradation has been demonstrated in several bacteria and this field is growing [11]. Recently, *Klebsiella pneumoniae* NX-1, *Pseudomonas putida* NX-1 and *Ochrobactrum tritici* NX-1 strains have been isolated from leaf mould samples [12]. *Ochrobactrum oryzae* BP03 was recently isolated from wood-feeding termites' guts and shown to delignify wheat straw [13]. Proteomic analysis has been conducted on some lignin degrading strains including *Pandoraea* sp. ISTKB [14]. Lignin-degrading bacteria have also been analysed for use in product generation from black liquor [15].

If we assume that bacteria are everywhere [16] and readily able to colonize those resources they are able to metabolize, the dearth of lignin-degrading bacteria discovered to date in black liquor vats suggests that black liquor may simply be too recalcitrant to degrade biologically. Perhaps more lineages of bacteria will evolve the ability to degrade black liquor quickly in alkaline baths, but just as likely they will not. Recently, however, a growing body of literature has begun to suggest that while many bacterial species are, indeed, everywhere their food can be found, some bacterial species are restricted to geographic areas or habitats and are relatively dispersal limited. Such is the case, for example, with some deep sea vent associated bacteria (e.g. *Sulfolobus* [17]) and may well be common among species with narrow habitat requirements [18]. We hypothesize that restricted habitats and their species may be promising, and poorly characterized, sources of novel enzymes, including those able to both degrade lignin and to do so under alkaline conditions [18].

One type of habitat that regularly harbours unique bacterial flora is the gut of insects [19,20]. Insect guts are miniature bioreactors, and insects have evolved diverse gut morphologies and physiologies in partnership with specialized microbial floras. Together, these features facilitate insects' ability to digest a diversity of unusual foods as well as to defend against pathogens [20]. We predicted, *a priori*, that the insects most likely to host lignin-degrading bacteria were likely to be those that feed on food resources (1) that contain recalcitrant carbon compounds and (2) for which the flux (availability per unit time) of those compounds is low, such that selection should maximize the completeness of degradation to give the highest nutritional benefit from scarce resources. Many detritivorous insects feed on relatively recalcitrant compounds (including, for example, thousands of species of termites and collembola), but in most cases such compounds are relatively abundant (e.g. all the dead leaves and wood of a forest). However, for organisms living in caves, and cavelike habitats such as nests,

cellars and basements, food resources are often dominated by relatively recalcitrant carbon compounds are scarce. It is this carbon scarcity that leads selection to favour, for example, cave-adapted organisms with exoskeletons that are relatively thin and transparent. Similarly, in such habitats selection should favour insects with gut microbes able to degrade recalcitrant carbon compounds, including lignin. But even if gut microbes from insects from caves, nests and basements are able to degrade lignin, they might not be able to do so under alkaline conditions. We hypothesize that the subset of insects likely to have gut microbes able to degrade lignin and to do so in alkaline conditions are likely to include species that both live in cavelike habitats and have simple guts. Termites, as the counter example, have complex, compartmentalized guts in which microbes are exposed to relatively invariant conditions. Species with simple guts create conditions for microbes that vary as a function of what they have eaten and even the stage of digestion, conditions that might range from relatively alkaline to relatively acidic.

Here, we use this framework to conduct a targeted search for bacterial species that can digest the lignin present in black liquor. We focus our efforts on two insect species (the greenhouse camel cricket, *Diestrammena asynamora* and the hide beetle, *Dermestes maculatus*), both of which are commonly associated with human homes as well as caves and nests, respectively. Human homes, like caves, have relatively scarce resources that often contain high levels of recalcitrant carbon compounds. Our interest in *D. asynamora* was first piqued during a citizen science survey we conducted about the animals that live in their houses. One surprising result of this survey was that greenhouse camel crickets (Rhaphidophoridae) were an extremely common presence in homes over many parts of the USA [21]. Surprisingly, the distribution of camel crickets reported by participants did not match well with the known distribution of camel crickets based on previous work [22]. When we followed up this survey with a call for citizen scientists to submit photographs and specimens of camel crickets in their homes, we found that the majority of camel crickets in homes across North America were in fact a non-native species from temperate Asia (*D. asynamora*) that was known to be a pest in greenhouses but not known in homes. We estimate these camel crickets may now number in the tens of millions [22]. The responses from citizen scientists also led us to identify a second species, the Japanese camel cricket (*Diestrammena japanica*) [21], which had not previously been documented in the literature as being present in the USA. We reported on this discovery, both in a scientific paper and to the public, and were quickly flooded with responses which included new observations about camel crickets, poems and pictures, and then questions [21]. One of the most common questions was 'what good are they anyway?' Although initially a frustrating response, it was this question from the public that led us to consider more specifically whether their gut microbes might be able to live in and degrade black liquor. In addition to *D. asynamora*, we conducted a parallel investigation on another common insect of human houses, the hide beetle, *Dermestes maculatus* (Dermestidae). Species of hide beetles are the most abundant insects in human homes and like greenhouse camel crickets rely on foods rich in lignocellulose [23,24].

We carried out several steps to consider whether gut microflora of greenhouse camel crickets and hide beetles might include bacteria able to degrade black liquor. We surface sterilized greenhouse camel crickets and hide beetles so as to kill bacteria on their exoskeletons and focus on gut-associated microbes. We then homogenized the greenhouse camel crickets and hide beetles and plated the homogenate on agar plates that contained dilute black liquor (so as to select for those bacteria able to live in black liquor). We then replated each of the resulting colonies on a series of substrates representing the components of black liquor in isolation and at various concentrations. We found isolates able to grow on all of the substrates included in the screen. We identified the isolates and then focused in on the species that appeared best able to grow in black liquor, *Cedecea lapagei*. We sequenced the entire genome of *C. lapagei* and then, having done so, were interested in determining what metabolic capabilities and specific genes may enable this facile bacterium to grow on black liquor and lignocellulose components.

## 2. Methods

### 2.1. Screening and isolation of bacteria from greenhouse camel crickets

A total of 158 Asian camel crickets (*Diestrammena asynamora*) were collected from seven houses that we had identified based on our earlier studies as having been greenhouse camel crickets. These greenhouse camel crickets were frozen at −20°C for initial storage. Hide beetles (*Dermestes maculatus*) were obtained from the North Carolina Museum of Natural History and similarly frozen for initial storage (Raleigh,

North Carolina). Later, three samples of each species were surface sterilized using the method in Arnold & Herre [25]. A whole cricket or beetle was placed in 70% ethanol for 2 min and then washed with 0.5% NaOCl for 2 min followed by a wash in sterile ice-cold 0.85% NaCl. Each insect was then homogenized using a sterile mortar and pestle in 1 ml of sterile ice-cold 0.85% NaCl. A total of three greenhouse camel crickets and three hide beetles were used in this study, each sterilized, homogenized and plated separately.

Subsamples of the homogenates from each insect were spread separately onto three 1% black liquor M9 minimal media agar plates and incubated at 25°C. Black liquor was obtained from the Department of Forest Biomaterials at North Carolina State University with pH 12.7, and 13.7% solids [26]. Plates were inspected daily for growth. Colonies from homogenate plates were selected and streaked onto 1% black liquor M9 minimal media agar and incubated at 25°C for 24–48 h (until bacterial growth was visible) to visually inspect for pure culture. We predicted that bacteria able to grow on this medium would be the subset of taxa from greenhouse camel crickets and hide beetles able to survive in the presence of black liquor and perhaps able to metabolize some of its components. M9 minimal medium was composed of 0.04 M $Na_2HPO_4$, 0.02 M $KH_2PO_4$, 18 mM $NH_4Cl$, 8.6 mM NaCl, 27 µM $CaCl_2$, 1 mM $MgSO_4$, according to Lech & Brent [27]. Luria-Bertani (LB) medium was prepared according to Lech & Brent [27] and included 10 g tryptone, 5 g yeast extract, and 5 g NaCl ($l^{-1}$ of distilled water).

From among the bacterial colonies that grew on the M9 media with dilute black liquor we sought to identify the subset of strains able to grow on each component of lignocellulose that is present in black liquor (lignin, hemicellulose (xylan) and cellulose). We plated each colony onto a selection of minimal media that had the lignocellulose components as the sole source of carbon. In addition, the bacteria were streaked onto Congo Red media at pH 4, 7 and 10 made according to Gupta et al. [28] in order to determine whether the isolates could degrade cellulose and grow at a range of pH. Bacteria were also streaked onto M9 minimal media with 0.2% of the following carbon sources: 1% black liquor (pH 11.5), 5% black liquor (pH 12), 10% black liquor (pH 12.3), hardwood flour (pH 6), softwood flour (pH 6), Biochoice lignin (BCL) (pH 5), carboxylmethyl cellulose (CMC) (pH 6), beechwood xylan (pH 7), xylose (pH 7) or pectin (pH 7) to assess whether the isolates could metabolize the various lignocellulose sources and components.

The carbon sources used for media preparation were obtained from the following sources: BCL from Domtar Inc., CMC from Alfa Aesar, beechwood xylan from Sigma-Aldrich, and xylose and pectin from Acros Organics. Wood flour was produced from birch (hardwood) species and pine (softwood) chips by milling to 0.2 mm particles [29]. Congo Red and bacteriological-grade agar were sourced from Amresco, yeast extract from Beckton Dickenson and tryptone from Thermo Fisher Scientific. Inorganic media components were purchased from Amresco (potassium phosphate monobasic anhydrous, and magnesium sulfate heptahydrate), Thermo Fisher Scientific (sodium phosphate dibasic anhydrous, calcium chloride, sodium chloride and sodium hydroxide) and BDH (ammonium chloride).

Growth on these agar media was inspected visually daily. Colonies from the initial insect homogenates grew slowly. Growth was apparent after several weeks.

## 2.2. Bacterial identification

Bacterial isolates capable of growth on up to 10% black liquor were sequenced to determine identity. Cells from 5 ml cultures of the isolated strains grown in LB media for 24 h were collected by centrifugation (13 000g for 5 min). Genomic DNA was isolated from this cell pellet using the Qiagen Gentra Puregene Bacteria Kit (CA) and visualized by electrophoresis on a 1% agarose gel. Amplification of the 16S rRNA by polymerase chain reaction (PCR) was accomplished using universal primers 8F (5′- AGAGTTTGATCCTGGC TCAG-3′) and 1492R (5′-GGTTACCTTGTTACGACTT-3′) [30]. The PCR product was electrophoresed through a 1% agarose gel for visual inspection and then subsequently prepared for sequencing using the Qiagen PCR clean-up kit (CA). Sequencing was performed by Eton Bioscience (Durham, NC) using 8F and 1492R primers. Sequencing results were analysed using the Basic Local Alignment Search Tool (BLAST) [31].

## 2.3. Genomic DNA isolation, sequencing and assembly from *C. lapagei*

Based on the results of the culture-based analysis, one species, *Cedecea lapagei*, was chosen for further analyses. Genomic DNA of C. lapagei was extracted using the Qiagen Yeast/Bact. Kit. C. lapagei genomic DNA was submitted to the Duke University Sequencing and Genomic Technologies Resource

for PacBio SMRT sequencing (RSII platform). A DNA library was prepared using one SMRT cell, resulting in 58 551 raw reads with a mean read length of 11 412 bases, totalling 668 190 553 bases. Generated reads were introduced into the Hierarchical Genome Assembly Process (HGAP), assembled with the Celera Assembler and polished with Quiver. Resulting assemblies produced two contigs with a total genome size of 4 691 775 bp. The 4.6 Mb genome has a total GC content of 55% with 112 RNAs (rRNAs and tRNAs) [32]. Annotation was performed using Rapid Annotation using Subsystem Technology (RAST) [33]. RAST predicted 4287 coding sequences for the genome of *C. lapagei*. The RAST server was used to classify functional groups of the coding sequences (CDSs) and construct a metabolic model. This whole-genome shotgun project has been deposited at DDBJ/EMBL/GenBank under the accession no. PIQM00000000. The predicted proteins were used to create a metabolic map using KASS [34].

## 2.4. Laccase activity of *C. lapagei*

*Cedecea lapagei* was cultured in LB broth, lennox (Fisher) with or without 1% BCL for 48 h. Cultures reached late-log phase with an $OD_{600}$ 2.08 for LB broth and 1.00 for LB broth with 1% BCL. Cells were centrifuged at 4300$g$ for 30 min. The supernatant was collected. Protein concentration of the supernatant was determined using the Peirce BCA protein assay kit (Fisher Scientific) according to the manufacturer's instructions. Laccase activity was determined as outlined in Huang *et al.* [35]. Supernatant, 50 µl, was added to a 96 well-plate. A freshly made solution of 3 mM 2,2′-azino-bis(3-ethyl-benzothiazoline-6-sulfonic acid) (ABTS) (Sigma) in McIlvaine buffer pH 3, 4, 5, 6, 7 or 8 was added to the plate (150 µl). The absorbance was read at 420 nM for 10 min using a BioTek Synergy HTX microplate spectrophotometer. One enzyme unit (U) is defined as the amount of enzyme that oxidizes 1 µmol of substrate per minute. The purified laccase enzyme from *Trametes versicolor* (Sigma) was used as a positive control.

## 2.5. Secretome analysis of *C. lapagei*

The putative proteins were analysed for secretion using PSORTb (http://www.psort.org/psortb/) because it has high precision (96.5%), allows multiple sequences to be analysed at once, and differentiates between Gram positive and Gram negative bacteria [36]. Overnight cultures of *C. lapegi* in stationary phase were inoculated at 1% (v/v) into M9 minimal media with glucose, CMC, beechwood xylan or BCL at the sole carbon source at 0.2%. Cultures were incubated at 37°C with shaking for 5 d. The secretomes were collected after bacterial growth by centrifugation at 4700$g$ for 10 min. The resulting supernatant was filter sterilized using a 0.2 µm syringe filter. The supernatant was concentrated on Vivacon 10 kDa spin columns. The proteins concentrated on the filter of these columns were then denatured and digested with trypsin following the filter-aided sample preparation (FASP) protocol of Wisniewski *et al.* [37]. The proteins on the filter were reduced with 10 mM DTT at 50°C for 30 min, alkylated with 30 mM iodoacetamide for 30 min at room temperature in the dark. The protein was digested with trypsin overnight at 37°C using a 1 : 20 trypsin : protein ratio. Peptides were stored at −20°C until use. Secreted proteins produced by these bacteria were analysed by nanoscale LC/MS/MS using an Easy nLC 1000 liquid chromatograph coupled to an Orbitrap Elite mass spectrometer system to measure differences in secretome protein profiles relative to cultures grown on glucose. Proteins were identified by searching peptide product ion spectra against a *C. lapagei* protein sequence using Proteome Disover (Thermo), and database using Mascot (Matrix Science). Differences in protein profiles between carbon sources were analysed using Scaffold (Proteome Software). Details of the LC/MS/MS including data acquisition and post-acquisition analysis appear in the electronic supplementary methods.

## 3. Results

From the three *D. asynamora* specimens sampled, we screened 25 bacterial colonies for growth on lignocellulosic components. Seventeen isolates were capable of growth on up to 10% black liquor and were then sequenced. Nine different bacterial strains were identified (table 1). From the three *D. maculatus* beetles that were sampled, 25 bacterial colonies were also screened for growth on lignocellulosic compounds, resulting in five different bacterial strains that were isolated and identified (table 2).

**Table 1.** Bacterial species from greenhouse camel crickets scored for growth on each of the substrates considered. Isolate identities were determined using NCBI BLAST. M9 (M9 minimal media), BL (black liquor), HWf (hardwood flour), SWf (softwood flour), Lig (lignin), CMC (carboxymethylcellulose), Beech (beechwood xylan), Xyl (xylose), Pect (pectin). (+) indicates growth on agar plates, (+/−) indicates indeterminate growth on agar plates, (−) indicates no growth on agar plates.

| 16S rRNA identity | % match | LB CONGO red | | | M9 BL | | | M9 0.2% carbon source | | | | | | |
|---|---|---|---|---|---|---|---|---|---|---|---|---|---|---|
| | | pH 4 | pH 7 | pH 10 | 1% | 5% | 10% | HWf | SWf | Lig | CMC | Beech | Xyl | Pect |
| Advenella mimigardefordensis strain DPN7 | 100 | − | + | − | + | + | + | + | + | − | + | + | + | + |
| Alcaligenes faecalis strain NBRC 13111 | 98 | − | + | + | + | + | + | + | + | − | + | + | + | + |
| Cedecea lapagei strain DSM 4587 | 96 | + | + | + | + | + | + | + | + | + | + | + | − | + |
| Enterobacter cloacae strain ATCC 13047 | 98 | + | + | − | + | + | + | + | + | − | + | + | + | + |
| Escherichia sp. UIWRF0668 | 90 | + | + | + | + | + | + | + | + | − | + | + | + | + |
| Hafnia alvei strain JCM 1666 | 100 | + | + | + | + | + | + | + | + | − | + | + | + | − |
| Hafnia alvei strain JCM 1666 | 99 | + | + | + | + | + | − | + | − | − | − | − | + | + |
| Hafnia alvei strain JCM 1666 | 99 | + | + | − | + | + | + | + | + | − | + | + | − | − |
| Hafnia alvei strain JCM 1666 | 99 | + | + | + | + | + | + | − | − | − | − | − | + | + |
| Hafnia alvei strain JCM 1666 | 100 | + | + | − | + | + | + | + | − | − | − | − | + | + |
| Hafnia alvei strain JCM 1666 | 99 | + | + | − | + | + | + | − | − | − | − | − | + | + |
| Hafnia alvei strain JCM 1666 | 99 | + | + | − | + | + | + | + | − | − | + | + | + | + |
| Hafnia paralvei strain ATCC 29927 | 99 | + | + | + | + | + | − | + | − | − | + | + | + | + |
| Obesumbacterium proteus strain NCIMB 8771 | 99 | + | + | − | + | + | − | + | − | − | + | − | + | − |
| Obesumbacterium proteus strain NCIMB 8771 | 99 | + | + | − | + | + | + | + | + | − | + | + | + | + |
| Obesumbacterium proteus strain NCIMB 8771 | 99 | + | + | − | + | + | + | + | − | − | − | − | + | − |
| Yersinia massiliensis strain 50640 | 99 | + | + | + | + | + | + | − | − | − | − | − | + | + |

**Table 2.** Bacterial species from hide beetles scored for growth on each of the substrates considered. Symbols and data are the same as given in table 1.

| 16S rRNA Identity | % match | LB congo red | | | M9 BL | | | M9 0.2% carbon source | | | | | | |
|---|---|---|---|---|---|---|---|---|---|---|---|---|---|---|
| | | pH 4 | pH 7 | pH 10 | 1% | 5% | 10% | HWf | SWf | Lig | CMC | Beech | Xyl | Pect |
| *Gibbsiella greigii* strain FRB 224 | 99 | + | + | + | + | + | + | − | + | − | − | + | + | + |
| *Proteus mirabilis* strain ATCC 29906 | 100 | − | + | + | + | + | + | + | − | − | + | − | + | − |
| *Proteus mirabilis* strain ATCC 29906 | 100 | − | + | + | + | + | + | − | − | + | − | − | + | − |
| *Proteus mirabilis* strain ATCC 29906 | 99 | − | + | + | + | + | + | − | − | − | − | − | + | − |
| *Proteus mirabilis* strain HI4320 | 99 | + | + | + | + | + | + | − | + | − | + | + | + | + |
| *Proteus mirabilis* strain HI4320 | 99 | + | + | + | + | + | + | + | + | + | − | + | + | + |
| *Proteus mirabilis* strain HI4320 | 100 | + | + | + | + | + | + | − | + | − | + | + | + | − |
| *Proteus mirabilis* strain ATCC 29906 | 100 | − | + | + | + | + | + | − | − | − | + | − | + | − |
| *Proteus mirabilis* strain ATCC 29906 | 99 | +/− | + | + | + | + | + | + | + | − | + | + | + | + |
| *Proteus mirabilis* strain ATCC 29906 | 99 | + | + | + | + | + | + | + | − | − | − | + | + | +/− |
| *Proteus mirabilis* strain ATCC 29906 | 99 | + | + | + | + | + | + | + | − | − | − | + | + | − |
| *Proteus mirabilis* strain ATCC 29906 | 99 | + | + | + | + | + | + | + | − | + | + | + | + | + |
| *Proteus mirabilis* strain ATCC 29906 | 100 | +/− | + | + | + | + | + | − | − | − | − | − | + | − |
| *Proteus mirabilis* strain ATCC 29906 | 99 | − | + | + | + | + | + | − | − | − | − | − | + | − |
| *Raoultella planticola* strain DSM 3069 | 99 | + | + | + | + | + | + | + | + | + | + | + | + | + |
| *Xenorhabdus hominickii* | 99 | + | + | + | + | + | + | + | +/− | − | + | + | + | + |

**Table 3.** Number of subsystem features of the *C. lapagei* genome.

| functional category | no. sequences |
| --- | --- |
| cofactors, vitamins, prosthetic groups, pigments | 267 |
| cell wall and capsule | 224 |
| virulence, disease and defense | 114 |
| potassium metabolism | 32 |
| photosynthesis | 0 |
| miscellaneous | 50 |
| phages, prophages, transposable elements, plasmids | 42 |
| membrane transport | 227 |
| iron acquisition and metabolism | 46 |
| RNA metabolism | 232 |
| nucleosides and nucleotides | 122 |
| protein metabolism | 297 |
| cell division and cell cycle | 42 |
| motility and chemotaxis | 148 |
| regulation and cell signalling | 141 |
| secondary metabolism | 5 |
| DNA metabolism | 110 |
| fatty acids, lipids and isoprenoids | 138 |
| nitrogen metabolism | 40 |
| dormancy and sporulation | 5 |
| respiration | 153 |
| stress response | 186 |
| metabolism of aromatic compounds | 18 |
| amino acids and derivatives | 464 |
| sulfur metabolism | 65 |
| phosphorus metabolism | 51 |
| carbohydrates | 633 |
| total | 3852 |

Although all the bacteria isolated through our screens shared the ability to grow on carbon substrates of interest, they differed with regard to the pH range in which they were able to grow and their relative ability to grow on the various carbon sources. In general, the bacteria we cultured were tolerant of a wide range of pHs (tables 1 and 2). Of the strains isolated from greenhouse camel crickets, all (17) were able to grow at neutral pH. Fifteen of the isolates were also able to grow at acidic pH (pH 4) and 9 at alkaline pH (pH 10). Most of the hide beetle isolates were able to grow at pH 4, 7 and 10, and thus are tolerant of a wide range of pHs. The exception was *Proteus mirabilis*, which was not able to grow under acidic conditions (pH 4).

As might be expected given that our initial selection media was M9 with 10% black liquor, all of the bacterial strains from the greenhouse camel cricket and hide beetle homogenates were able to grow on M9 with low concentrations of black liquor added. Three species were no longer able to grow once black liquor concentrations were increased to 10%, even though nutrients were still available in the M9 agar. At least one of the bacteria species was able to grow on hardwood, softwood, beechwood xylan or the compounds found in black liquor: pectin, xylose, CMC and lignin. Only two species, *C. lapagei* and *P. mirabilis*, were able to grow on lignin (BCL) as a sole carbon source. BCL is the precipitated kraft lignin from the black liquor produced in paper manufacturing from Plymouth Mill of Domtar Inc. It has been characterized by Hu *et al*. [38]. We chose to focus on *C. lapagei* for

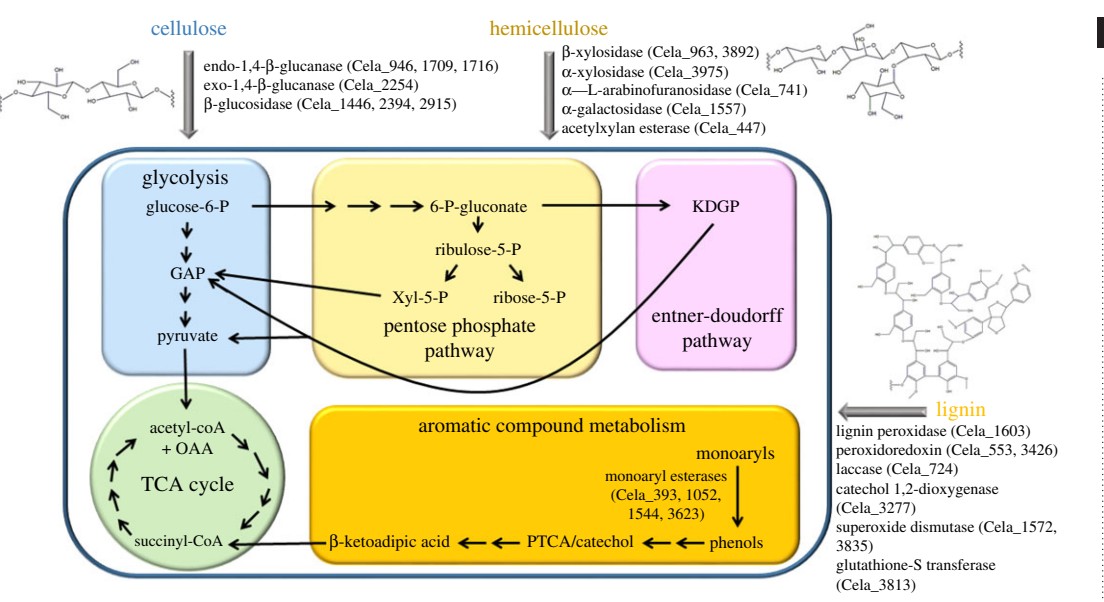

**Figure 1.** Metabolic model of lignocellulose degradation pathways in *C. lapagei* based on genome sequence. KEGG Model was created based on the data provided by annotation using KASS.

subsequent analyses because of its ability to grow under both acidic and alkaline conditions and because it can grow well on all of the lignocellulose sources and components tested other than the monomeric sugar xylose.

*Cedecea lapagei* has been poorly studied. It is a Gram negative, facultatively anaerobic, non-spore forming rod. The strain described in the literature grows from 15 to 37°C at pH 7 and 9, which is to say the strain we isolated from greenhouse camel crickets grows under both more basic and more acidic conditions than does the type strain [39]. The type strain of *C. lapagei* produces catalase and nitrate reductase enzymes as well as acidic metabolic products from the degradation of cellobiose, maltose, salicin and trehalose. It is also able to use cellobiose, galactose, galacturonate, glucose, malate, mannitol, mannose, salicin or trehalose as its sole carbon and energy source. *Cedecea lapagei* was first isolated from clinical sources in North America, though it is common in mosquitoes [40] and may also be present in other insects.

The *C. lapagei* genome was sequenced using PacBio large insert library preparation at the Duke University Genome Sequencing Shared Resource (DUGSSR). Assembly resulted in two large contigs of 4.6 Mb that were then submitted to RAST for annotation. The classification of the CDSs into functional categories is summarized in table 3. The KEGG metabolic map as determined by genome sequencing for lignocellulose metabolism is provided in figure 1. The secreted proteins (secretome) of *C. lapagei* were of special interest because they are most likely involved in the breakdown of black liquor waste components, as determined by growth on black liquor as the sole carbon source. Cellulose, hemicellulose, and lignin are large polymers that cannot directly be taken into the bacterium. Instead extracellular enzymes are made that break down these polysaccharides from cellulose and hemicellulose and polyaromatic compounds from lignin into smaller sugars or aromatics that can then be further metabolized inside the bacterium. pSortB was used to identify potential secreted protein-encoding genes from the annotation results. RAST predicted a total of 4287 genes of which 19 are predicted to be extracellular, 55 are outer membrane proteins and 152 have signal peptides but their location is unknown. This list of interesting enzyme candidates was narrowed down to include those with enzymes that may be involved in lignocellulose degradation by *C. lapagei* based on their predicted function (e.g. arabinofuranosidase, pectinesterase, chitinase, glucanase and copper oxidase enzymes, table 4).

The supernatant resulting from when *C. lapagei* was grown in the presence of BCL demonstrated an increase in absorbance at 420 nm when added to ABTS (figure 2). Laccase activity was detected in the supernatant of *C. lapagei* cultures whose media had a pH of 6–8 and were grown in the presence of lignin. The highest detected activity was 5 mU/µg when cultures were grown at pH 6. *C. lapagei* cultures grown on LB demonstrated laccase activity at pH 7 and 8. *T. versicolor* had optimal activity at pH 3 (1.24 U/µg).

**Table 4.** Secretome proteins that may play a role in degradation of lignocellulose Identified by LC-MS/MS.

| cellulose | hemicellulose | lignin | gene number | predicted location | function | reference |
|---|---|---|---|---|---|---|
| lignin breakdown/toxicity reduction | | | | | | |
| X | | | 3831 | cytoplasmic | lactoylglutathione lyase (EC 4.4.1.5) | Forsberg et al. [41] |
| X | | | 3299 | cytoplasmic | phenolic acid decarboxylase (EC 4.1.1.-) | Forsberg et al. [41] |
| X | | | 3813 | cytoplasmic | glutathione S-transferase (EC 2.5.1.18) | Forsberg et al. [41] |
| | X | X | 1873 | cytoplasmic | malonate decarboxylase α subunit | Dashtban et al. [42] |
| | X | X | 4142 | cytoplasmic | paraquat-inducible protein B | Steelink [43]; de Gonzalo et al. [44] |
| X | X | X | 3835 | cytoplasmic | superoxide dismutase [Fe] (EC 1.15.1.1) | Rashid et al. [45] |
| X | | | 2354 | cytoplasmic | folate-dependent protein for Fe/S cluster synthesis/repair in oxidative stress | de Gonzalo et al. [44] |
| X | | | 2359 | cytoplasmic | thiol:disulfide interchange protein DsbC | Dashtban et al. [42] |
| X | | | 1655 | cytoplasmic | homologue of E. coli Hemx protein | Stenberg et al. [46] |
| X | X | X | 887 | cytoplasmic | osmotically inducible protein OsmY | Hingston et al. [47] |
| X | | X | 3516 | unknown | molybdopterin oxidoreductase (EC 1.2.1.2) @ selenocysteine-containing | Dashtban et al. [42] |
| X | X | X | 173 | cytoplasmic | non-specific DNA-binding protein Dps/Iron-binding ferritin-like antioxidant protein/Ferroxidase (EC 1.16.3.1) | de Gonzalo et al. [44] |
| X | X | | 1572 | cytoplasmic | manganese superoxide dismutase (EC 1.15.1.1) | Neumann et al. [48]; Rashid et al. [45] |
| X | X | X | 3540 | cytoplasmic | osmotically inducible protein C | Hingston et al. [47] |
| X | X | X | 341 | cytoplasmic | alkyl hydroperoxide reductase protein C (EC 1.6.4.-) | Steelink [43] |
| X | X | X | 553 | cytoplasmic | alkyl hydroperoxide reductase subunit C-like protein | Steelink [43] |
| X | X | X | 3426 | cytoplasmic | thiol peroxidase, Tpx-type (EC 1.11.1.15) | Brown et al. [49] |
| X | X | X | 904 | cytoplasmic | organic hydroperoxide resistance protein | de Gonzalo et al. [44] |

(Continued.)

**Table 4.** (*Continued.*)

| cellulose | hemicellulose | lignin | gene number | predicted location | function | reference |
|---|---|---|---|---|---|---|
| X | X | X | 1639 | cytoplasmic | thioredoxin | Steelink [43] |
| X | X | | 4032 | cytoplasmic | 3-oxoacyl-[acyl-carrier protein] reductase (EC 1.1.1.100) (resistance to furfural) | Wierckx et al. [50] |
| X | X | | 1603 | cytoplasmic | catalase (EC 1.11.1.6)/peroxidase (EC 1.11.1.7) | Brown et al. [49] |
| X | | | 1732 | cytoplasmic | glutathione reductase (EC 1.8.1.7) | De Gonzalo et al. [44] |
| X | | | 2226 | cytoplasmic | methylglyoxal reductase, acetol producing (EC 1.1.1.-)/2,5-diketo-D-gluconate reductase A (EC 1.1.1.274) | Kersten & Cullen [51] |
| carbohydrate degradation/modification | | | | | | |
| X | | | 3929 | cytoplasmic | N,N'-diacetylchitobiose-specific 6-phospho- β-glucosidase (EC 3.2.1.86) | Horn et al. [52] |
| X | | | 3793 | cytoplasmic | mannose-6-phosphate isomerase (EC 5.3.1.8) | Mathews et al. [3] |
| X | | | 3012 | cytoplasmic | phosphomannomutase (EC 5.4.2.8) | Mathews et al. [3] |
| X | X | | 3253 | cytoplasmic | trehalase (EC 3.2.1.28); periplasmic trehalase precursor (EC 3.2.1.28) | Habe et al. [53] |
| X | X | X | 637 | cytoplasmic | phosphoheptose isomerase 1 (EC 5.3.1.-) | Horn et al. [52] |
| X | X | | 458 | periplasm | UDP-sugar hydrolase (EC 3.6.1.45); 5'-nucleotidase (EC 3.1.3.5) | Voigt et al. [54] |
| X | | | 1365 | cytoplasmic | mannitol-1-phosphate 5-dehydrogenase (EC 1.1.1.17) | Brown et al. [84] |
| X | X | X | 2895 | cytoplasmic | galactose/methyl galactoside ABC transport system, D-galactose-binding periplasmic protein MglB (TC 3.A.1.2.3) | Horn et al. [52] |
| X | X | X | 2729 | cytoplasmic | PTS system, glucose-specific IIA component (EC 2.7.1.69) | Horn et al. [52] |
| X | | X | 3933 | cytoplasmic | PTS system, N,N'-diacetylchitobiose-specific IIB component (EC 2.7.1.69) | Horn et al. [52] |
| X | X | X | 2024 | cytoplasmic | uncharacterized ABC transporter, auxiliary component YrbC | Horn et al. [52] |

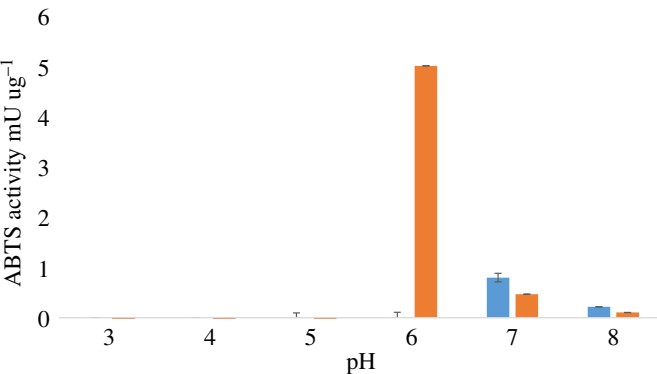

**Figure 2.** ABTS-oxidizing activity of *C. lapagei* supernatants grown in LB and LB with 1% BCL. Error bars indicate the standard error of the mean for biological triplicates.

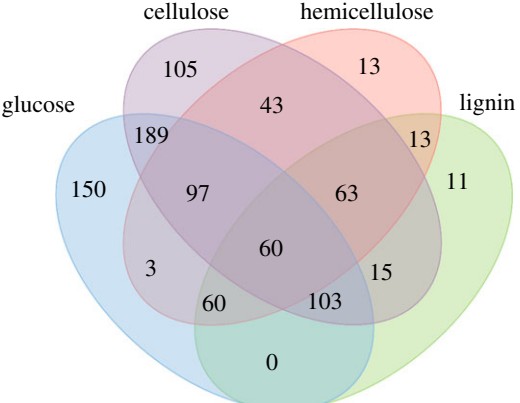

**Figure 3.** Comparison of unique proteins of the *C. lapagei* secretome present after growth on glucose, cellulose, hemicellulose, and lignin as the sole carbon sources. Numbers represent matches for peptides using the *C. lapagei* predicted amino acid sequences as the database.

After growth on lignocellulosic components as the sole carbon source, the secretome was enriched by centrifugation to remove the cells and retain the growth medium where the extracellular proteins are found. The proteins present were digested with trypsin and then identified by LC/MS/MS. These results identify proteins that are made when grown on carbon sources. Some secretome proteins were only found on one carbon source and therefore considered unique to that substrate (lignocellulose components cellulose, hemicellulose or lignin). There were 105 unique *C. lapagei* proteins in the secretome after growth on cellulose, 23 when grown on hemicellulose, and 20 after growth on lignin (figure 3). The list of identified secretome proteins is presented in electronic supplementary material, table S1. LC/MS/MS data show that *C. lapagei* is differentially producing proteins in response to growth on these carbon sources. A portion of these proteins is likely involved in degrading a particular substrate, and thus provide a more refined list of potential lignocellulose-degrading enzymes. Future work will focus on the recombinant expression and characterization of the enzymes involved in the degradation of cellulose, hemicellulose and lignin (identified by the presence in secretome) that could be used for industrial applications involving enzymatic breakdown of lignocellulose.

# 4. Discussion

Ecologists and conservation biologists have long argued that one of the key reasons for the conservation of biological diversity is to conserve the potential value that species might have for future generations [55]. Such potential value is great in no small part because it integrates discoveries to be made across tens, thousands or even more human generations (contingent on our persistence). Yet while scientists make these claims often, it is far rarer that they actually set out systematically to find novel uses of

species. Where such attempts have been made, they have tended to screen species randomly and focus on high-value uses (cancer drugs) and typically have met with little success [56]. In response to public comment on our research about greenhouse camel crickets, we decided to study the potential value of greenhouse camel crickets. Rather than randomly screening for potential uses, we employed ecological insights about the diets and life history of greenhouse camel crickets and other cave- and house-dwelling insects to inform the capabilities for which we searched; we focused on the potential of greenhouse camel crickets to host bacteria able to break down black liquor, the waste of the paper pulp industry and the lignin in black liquor. In doing so, we not only found bacteria able to carry out this process, but we identified three new bacterial species known to be able to break down lignin, as demonstrated by growth on lignin as the sole carbon source. We also identified nine new species able to use alkaline black liquor (as demonstrated by growth on black liquor as the sole carbon source), which contains lignin and sugars.

We are not the first to search for industrially useful bacteria in insects. Termites, for example, have long been a focus for sources of lignocellulose-degrading gut microorganisms and lignin-degrading bacteria [57,58]. However, our work demonstrates the value of considering additional insect taxa that digest lignocellulosic materials [59]. In doing so, we have identified five new bacterial strains capable of growth on lignin as the sole carbon source after considering just two insect species: one species of greenhouse camel cricket (*Diestrammena asynamora*) and one species of hide beetle (*Dermestes maculatus*). Before this work, there were only three bacterial species, *Novospingobium* sp. B-7, *Paenibacillus glucanolyticus* SLM1 and *Paenibacillus glucanolytius* 5162, capable of growth in lignin as the sole carbon source [60,61]. Further characterization of the ability of *C. lapagei, P. mirabilis* strain ATCC 29906, *P. mirabilis* strain HI4320, *R. planticola* and *X. hominickii* to grow on lignin is required for their application to industrial processes. Lignin degradation most likely occurs as a result of the action of secreted peroxidase or laccase enzymes and a suite of enzymes involved in the breakdown of low-molecular weight aromatics, but the optimal conditions required for this activity are still unknown [10]. It is also possible that these bacteria can grow on lignin-associated aromatics present in BCL media solutions used. In this case, other enzymes may be involved with breaking down these smaller molecular weight compounds [17]. Further research will focus on the chemical characterization on black liquor and lignin before and after growth of the isolated bacteria. Many bacteria found in the soil or in aquatic systems with wood and leaf debris may break down lignin or lignin fragments but require other carbon sources for this metabolism [59]. It is possible that many of the bacteria isolated from these insects are able to degrade lignin when other substrates can be used as carbon sources, such as those capable of growth on wood flour. Wood flour is a growth substrate that contains lignin as its main component in the presence of carbohydrate polymers.

*Cedecea lapagei* is a particularly interesting isolate from the greenhouse camel cricket. This bacterium is capable of growth at pH 4, 7 and 10 on a variety of different lignocellulosic compounds as the sole carbon source. This phenotype suggests that *C. lapagei* has a unique metabolism and is capable of producing enzymes of special interest that can transform lignocelluosic waste materials into component sugars and monoaromatics that can be used in high-value chemical production. Its sequenced genome was similar in size to other *Cedecea* strains. We found over 4000 predicted enzyme-encoding genes in the genome of *C. lapagei* isolated from greenhouse camel crickets. Typically, several types of enzymes are required to work in concert in order to break down the cellulose, hemicellulose and lignin components of lignocellulose, and our annotation of the *C. lapagei* genome identified putative cellulase, xylanase and family 65 glycosyl hydrolase that would likely function in lignocellulose degradation.

Analysis of the supernatant for laccase activity demonstrated that it had the ability to oxidize ABTS under laboratory conditions. Oxidation of ABTS by *C. lapagei* supernatant was considerably lower than that of the purified laccase from *T. versicolor*, however, that is not unexpected given that the *T. versicolor* enzyme is from a purified preparation, whereas the *C. lapagei* laccase is from a crude supernatant. Thus, these results suggest that the proteins secreted by this bacterium could participate in lignin degradation and bioconversion.

Analysis of a bacterial secretome is also a helpful tool for identifying proteins for particular applications such as lignocellulose degradation [54,62]. Secretome analysis may reveal genes that were overlooked by genomic analysis. Therefore, secretome analysis using LC/MS/MS was performed in an effort to identify additional potential enzymes required for lignocellulose metabolism by *C, lapagei*. The analysis was focused on proteins secreted by *C. lapgei* but resulted in only a limited number of enzymes with secretion signals. It is possible that proteins identified without secretion signals were cytosolic due to cell lysis during growth. It is also possible that secretion signals were not identified due to limitations of the bioinformatic approach used, as some proteins can play a role both in the cytoplasm and in the

secretome [62–64]. Intracellular and extracellular laccase enzymes have been found in bacteria such as *Azopsirillum lipoferum* and *Bacillus subtilis* [64,65]. The analysis revealed several unique proteins present in the secretome while growing on lignocellulose components that may be involved in the multi-enzyme pathway to degrade and grow on cellulose, hemicellulose or lignin as the sole carbon source (electronic supplementary material, table S1). Those enzymes with functions linked to lignocellulose degradation are listed in table 4. We found several proteins identified by LC/MS/MS that are involved in degradation, modification and transport of sugars, which make up cellulose and hemicellulose. Similar methods were used by Brown *et al*. [49] to identify proteins involved in lignocellulose degradation from *Amycolatopsis* sp. 75iv2 after growth on lignocellulose isolated from the grass *Miscanthus*. We have identified similar secreted proteins in this work that are involved in lignin and carbohydrate depolymerization and degradation (table 4; electronic supplementary material, table S2). While there is not a single defined pathway for bacterial lignin degradation, there have been several enzymes shown to function in lignin degradation, tolerance of lignin degradation and utilization of lignin-derived aromatic products from both fungal and bacterial sources [66–68]. Detoxification enzymes were identified in the secretome of *C. lapagei* when grown on all carbon courses. Superoxide dismutase, glutathione s-transferase, alkyl hydroperoxide reductase, glutathione reductase were enzymes identified when grown on media that contained cellulose or hemicellulose. Manganese superoxide dismutase from *Sphignobacterium* sp. T2 has been shown to degrade lignin [49]. Other enzymes that play a role in the protection of the bacterium from reactive oxygen species (ROS) like catalase and superoxide dismutase have also recently been implicated in lignin degradation. As an example, brown-rot fungi are known to be involved in the lignin depolymerization necessary to access cellulose, as a result of the production of Fenton-reactions which produce ROS that can cleave the aromatic subunits of lignin [45]. Metals are also involved in Fenton-reactions, and *C. lapagei* produces several metalloenzymes or proteins involved in metal resistance. In addition, it has become increasingly appreciated that tolerance to lignin-derived breakdown products is essential for microorganisms that degrade lignin. Lactoylglutathione lyase has recently been demonstrated to provide tolerance to the lignin breakdown product furfural [69]. Phenolic acid decarboxylase also plays a role in tolerance to the lignin monomer ferulic acid [69]. Peroxdiases have also been shown to play a role in oxidative damage formed during metabolism of lignin and other aromatic compounds as well as being differentially expressed by *Pandorawa* sp. ISTKB in the presence of lignin-derived aromatics [14]. The proteomic analysis of the secretome of *C. lapagei* presented here represents an excellent starting point for identifying and characterizing enzymes that could be developed for applications in the bioenergy, wood and pulping, textiles and chemical production industries requiring lignocellulose degradation and depolymerization enzymes.

One of the particularly interesting features of *C. lapagei* is the ability of this bacterium to grow in a broad pH range, including highly alkaline conditions. Bacteria tolerant of alkaline conditions are relatively rare, as are the conditions themselves, at least in nature. Soda lakes, such as Lake Natron in Tanzania Africa and Soda Dry Lake in California USA are some of the few naturally occurring environments that are strongly alkaline. Industrial activity, however, has created many additional alkaline environments. The ability of *C. lapagei* to grow at a high pH suggests that the enzymes it produces, especially those that are secreted, are likely to function at a high pH. Alkaliphilic enzymes have a variety of potential applications including their use in detergents, food processing and finishing of fabrics [70]. Just which of the enzymes secreted by *C. lapagei* will be useful for any particular process awaits better characterization of each enzyme and its fit to particular industrial needs.

The enzymes produced by *C. lapagei* about which we have the best understanding are those that are able to degrade lignocellulose. The lignocellulose-degrading enzymes produced by *C. lapagei*, as well as those produced by the other bacteria isolated from the greenhouse camel cricket (*Diestrammena asynamora*) and the hide beetle (*Dermestes maculatus*), have the potential to be directly applied to many industrial processes which include biofuel production, pulp and paper bleaching, bioremediation, textile dye decolorization and wine clarification. To date, attempts to carry out these processes using microbes have been hindered by a focus on fungi. However, fungal enzymes are difficult to produce in large quantities and are, with the exception of only a few alkaliphilic fungi (*Myrothecium verrucaria* and *Coprinus* sp.), not capable of degrading lignin at high pH. Bacterial enzymes have been viewed as the solution to overcome the problems associated with industrial applications of fungi, yet few bacterial, lignin-degrading enzymes have been characterized. Here we have identified a novel source of bacterial enzymes involved in lignin bioconversion. Notably the techniques described do not rely on sequence similarity to fungal lignin-degrading enzymes and therefore may identify novel enzymes (mechanism of action and/or enzymatic characteristics) from household insects.

The search for novel bacteria and enzymes has included screening private culture collections or efforts to search, species by species, for enzymes of value among samples from soil, lignin-containing waste or other habitats. It is hard to estimate the value of these screening techniques as companies are unlikely to publicize or publish unsuccessful attempts to find new useful bacteria or fungi [41]. We know more about the efficacy of attempts to search wild species, one by one. Here we propose a third approach, the search for useful species and enzymes based on ecology and natural history. This approach has been similar to those used to identify carbohydrate degrading enzymes from species such as the wood-boring marine isopod the gribble (*Limnoria quadripunctata*) and from the shipworm (*Lyroduc pedicellatus*) [71,72]. Because the greenhouse camel cricket (*Diestrammena asynamora*) feeds on resources that tend to include lignocellulose and be low in total flux, we hypothesized that it and other species with similar diets would be likely to host bacteria able to degrade lignocellulose. Both the greenhouse camel cricket and the hide beetle (*Dermestes maculatus*) indeed hosted such bacteria. Extending our hypothesis, it seems likely that additional lignocellulose-degrading species will be found in other arthropod species with low-carbon diets, such as many household and cave-dwelling species. Finally, based on an understanding of the general features of evolution and the tendency of traits to be phylogenetically constrained, we hypothesize that the many hundreds of other camel cricket and hide beetle species are likely to host additional bacteria able to break down lignocellulose. Such bacteria may produce enzymes that differ in perhaps subtle but functionally important ways.

We would be remiss if we concluded without considering the reason we did this work in the first place. We began to consider the potential value of the bacteria in the gut of greenhouse camel crickets because we were prompted to do so by the public. Hopefully, the answer is now clear. Along with their intrinsic value, these species possess bacteria with potentially enormous value to industry and society. It is easy for ecologists and evolutionary biologists to assume that someone else will apply the insights from these fields, whether to the discovery of useful organisms or some other application. But all too often the connections between basic ecologists and evolutionary biologists and applied biologists are limited, such that many of the ways that ecology and evolution might be useful in practical applications are never realized. Here, we show the value of a partnership between basic ecologists and evolutionary biologists, biochemists and applied microbiologists. The work we have done here is something the basic biologists among us would not have attempted if not prompted by the public. It is work we could not have accomplished without partnering with biochemists and applied microbiologists. Conversely, the organisms we targeted are species that are unlikely to have been targeted by the applied microbiologists without insights from ecology and natural history. We suspect this story is not unique to us, but rather is a general feature of the divide between basic ecology and evolutionary biology and application—a divide it would benefit us all to cross.

## 5. Conclusion

The results of a citizen science project revealed that greenhouse camel crickets (*Diestrammena asynamora*) were present in many homes in North America. Citizens who collected these data asked about the usefulness of this insect, and a team of ecologists, microbiologists and biochemists answered the question by using ecological theory to predict the types of potentially useful enzymes present in the insect flora. The greenhouse camel cricket and the hide beetle (*Dermestes maculatus*) are household insects that survive on recalcitrant carbon sources, and thus, the microflora of these insects were screened for growth on lignocellulosic compounds and the lignocellulosic industrial waste black liquor. From these two household insect species, 14 new strains of bacteria were isolated that are capable of growth on lignocellulose substrates. *Cedecea lapagei* was isolated from the greenhouse camel cricket and used for further study. The secretome of this bacterium after growth on lignocellulosic substrates was determined by LC/MS/MS to identify those enzymes that may be involved in lignocellulose degradation and therefore have potential for use in industrial processes that use this recalcitrant feedstock such as paper and pulp and biofuel production.

Data accessibility. Data are available from the Dryad Digital Repository: https://doi.org/10.5061/dryad.703h3c8 [73].
Authors' contributions. S.L.M. designed the experiments, collected data, analysed and interpreted data and drafted the manuscript; M.J.E. collected insects, selected appropriate method for insect sterilization and homogenization, and edited the manuscript; R.K.B. performed LC/MS/MS research, data collection and analysis as well as manuscript revision; M.B.G. designed LC/MS/MS experiments and edited the manuscript; A.M.G. designed the experiment and helped to draft the manuscript; R.R.D. conceived of the study, designed the experiment and helped to draft the manuscript. All authors gave final approval for publication.

Competing interests. We declare we have no competing interests.
Funding. NSF I Corps grant no. NSF1559771 for S.L.M., A.M.G. and R.R.D. and NSF 0953390 to RRD (which supported MJE). NSF DBI-1126244 and North Carolina Agricultural Research Service for R.K.B. and M.B.G.
Acknowledgements. We thank the Dunn, Grunden and Goshe labs at North Carolina State University.

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
