## [Reviewer comments · Royal Society Open Science]

Review History

RSOS-180748.R0 (Original submission)

Review form: Reviewer 1 (Hidetoshi Okuyama)

Is the manuscript scientifically sound in its present form?

Yes

Are the interpretations and conclusions justified by the results?

Yes

Is the language acceptable?

Yes

Is it clear how to access all supporting data?

Yes

Do you have any ethical concerns with this paper?

No

Have you any concerns about statistical analyses in this paper?

I do not feel qualified to assess the statistics

Recommendation?

Accept with minor revision (please list in comments)

Comments to the Author(s)

The manuscript is quite well written and almost no problems are found. But the size of letters used in Figures 1 and 2 is too small. The configurations of some molecules shown in Figure 1 are also too small.

Hidetoshi Okuyama, Formerly Hokkaido University.

Review form: Reviewer 2

Is the manuscript scientifically sound in its present form?

No

Are the interpretations and conclusions justified by the results?

No

Is the language acceptable?

Yes

Is it clear how to access all supporting data?

Yes

Do you have any ethical concerns with this paper?

No

Have you any concerns about statistical analyses in this paper?

No

Recommendation?

Major revision is needed (please make suggestions in comments)

Comments to the Author(s)

The manuscript 'Public questions spur the discovery of new bacterial species able to degrade lignin in industrial waste' presents a new source of bacteria and, potentially enzymes, that may be involved in lignocellulose degradation. I think this work is interesting from an ecological and evolutive point of view. However, it is very basic from the microbial point of view and in the lignin depolymerization field. The work is well written but there are many statements that would need to be toned down, modified, or eliminated regarding the discovery of enzymes or the novelty of the work. The secretome results also need to be more critically discussed. During the last years, there has been a growing number of manuscripts on the field of bacterial lignin degradation that would need to be reviewed to present a more realistic discussion.

Specifics:

- The introduction is lacking information on what it is known regarding lignin depolymerization

by bacteria. Please, include a paragraph about that. Below some key publications on this topic that contain a good source of references.

1. Huang XF et al, 2013. Isolation and characterization of lignin-degrading bacteria from rainforest soils. *Biotechnol Bioen*.
2. Linger et al, 2014. Lignin valorization through integrated biological funneling and chemical catalysis. *PNAS*.
3. Salvachua et al, 2015. Towards lignin consolidated bioprocessing: simultaneous lignin depolymerization and product generation by bacteria. *Green Chemistry*.
4. Lin et al. 2016 Systems biology-guided biodesign of consolidated lignin conversion. *Green Chemistry*.

- Page 5-line 7: what is exactly 'Biochoice lignin'? Is that a soluble or insoluble lignin stream?
 - Page 5-line 6: Did the authors try higher black liquor concentrations with the bacteria selected?
 - Considering that the paper is mostly focused in black liquor, a detailed composition of this substrate should be given. For instance, many black liquor streams contain carboxylic acids (i.e. acetate, glycolic acid, etc) and monomeric aromatic compounds (i.e. guaiacol, p-coumaric acid, ferulic acid, vanillic acid, etc) that can be utilized by bacteria as only carbon source. If this is the case, authors may be finding aromatic-catabolic bacteria and not lignin-degrading bacteria. Please present a table with the composition of the black liquor.

- Page 6, line 11. 'Turbid' is not a very precise way to present bacterial growth. If possible, present the values corresponding to optical density measurements (i.e. optical density at 600 nm).

- Page 6-line 45. It is very interesting (and not typical) to find bacteria that are able to grow at a such large range of pHs (4 to 10). Was the pH verified in the plates? It is worth considering that the buffering capacity of phosphate buffer (in M9) is between 6 and 8. Thus, different buffers might be preferred for pHs out of that range.

- Page 7-line 10. '... lignin as sole carbon source'. How was the lignin sterilized? Some sterilization processes (i.e autoclave) can release aromatic compounds from high molecular weight lignin and be utilized as carbon source by the bacteria. If this is the case, again, we cannot talk about lignin degradation only.

- Page 7-line 31- 'involved in the breakdown of black liquor'. To conclude this, the characterization of the black liquor (before and after the bacterial treatment) is needed. Again, were there aromatic compounds or other carbon sources apart from high molecular weight lignin? What was the lignin content? What was the size distribution of the lignin? If none of these parameters are given, sentences like this are recommended to be removed along the manuscript.

- Page 7, line 41, Table 4. There is a high number of cytoplasmic proteins in the secretome. Could that be just an effect of bacterial lysis and not secretion? Please comment on this.

- Page 8- line 27-line 31... It is highly recommended to further review the literature in this topic. There is a considerable number of bacteria able to utilize black liquors, kraft lignin, and the aromatic compounds on it. In fact, some of those bacteria are able to break down the lignin, which is not demonstrated in this work (see references above).

- Page 8, line 37. 'lignin as the sole carbon source'. Characterization is needed.

- Page 9-line 40. 'Brown rot fungi are known to hydrolyze lignin...'. Brown rot fungi do not break down lignin, they only modify it to attack the cellulose, that is why the residue after the fungal attack is brown (compared to white-rot fungi, which really degrade lignin). Please read book chapter from Barry Goodell entitled 'Brown-Rot Fungal Degradation of Wood: Our Evolving View'.

- Page 9-line 37. Authors present manganese superoxide dismutase as a potential enzyme involved in lignin depolymerization. Looking at Table 4, that enzyme was not even found in the lignin media. Discuss this critically. In addition, manganese superoxide dismutase is presented as a cytoplasmic protein. Lignin depolymerization is an extracellular process. Please comment on this as well.

- Table 4. Additional comments: there is another superoxide dismutase that appears in all media (also in lignin, 3835). This result suggests that this enzyme may not be involved in lignin break

down and only be related with detoxification. Authors also include in Figure 1 a Glutathione-S-transferase (3813) as potential enzyme in the lignin degradation pathway. However, this enzyme was not found either in the secretomic study in the lignin media (only in cellulose media). Typically, differential proteomic analyses give insights of the enzymes that are differentially expressed and needed to metabolize various substrates. However, these results are not conclusive and not critically discussed. Please consider all these comments to re-distribute the enzymes in the table (different sections) and be more critical in the discussion.

-Page 10-line 29- 'Here we have identified a novel source of bacterial enzymes involved in lignin degradation'. This statement is not correct until it is demonstrated (i.e. lignin characterization or enzymes assays).

-Page 10-line 35. Microbiologists are routinely looking into new sources of microorganisms and enzymes in different environments through bioprospecting. This is a very typical tool that this community uses and specifies in their papers. Saying that scientists 'focus on screening private culture collections' is not right. Please tone down. See references above and the one below.

-Page 10- line 40-41. The search of enzymes in insects or crustaceans has been reported years ago. Here just an example on a crustacean, the gribble, studied because of the habitat that colonized and in which powerful cellulases were discovered. Paper: King et al, 2010. Molecular insight into lignocellulose digestion by a marine isopod in the absence of gut microbes. PNAS.

-Page 10, line 1-4. Authors need to demonstrate lignin break down to make this statement.

Decision letter (RSOS-180748.R0)

08-Nov-2018

Dear Dr Mathews,

The editors assigned to your paper ("Public Questions Spur the Discovery of New Bacterial Species Able to Degrade Lignin in Industrial Waste") have now received comments from reviewers. We would like you to revise your paper in accordance with the referee and Associate Editor suggestions which can be found below (not including confidential reports to the Editor). Please note this decision does not guarantee eventual acceptance.

Please submit a copy of your revised paper before 01-Dec-2018. Please note that the revision deadline will expire at 00.00am on this date. If we do not hear from you within this time then it will be assumed that the paper has been withdrawn. In exceptional circumstances, extensions may be possible if agreed with the Editorial Office in advance. We do not allow multiple rounds of revision so we urge you to make every effort to fully address all of the comments at this stage. If deemed necessary by the Editors, your manuscript will be sent back to one or more of the original reviewers for assessment. If the original reviewers are not available, we may invite new reviewers.

When submitting your revised manuscript, you must respond to the comments made by the referees and upload a file "Response to Referees". Please use this to document how you have

responded to the comments, and the adjustments you have made. In order to expedite the processing of the revised manuscript, please be as specific as possible in your response.

Please note that rkblackb@ncsu.edu appears to have identified ScholarOne emails as spam, and will not be able to receive this message. Please ensure this author is forwarded a copy of the decision and asked to resolve the blacklisting of ScholarOne (this may require interaction with ScholarOne and their institution).

- Data accessibility

If you wish to submit your supporting data or code to Dryad (<http://datadryad.org/>), or modify your current submission to dryad, please use the following link:
<http://datadryad.org/submit?journalID=RSOS&manu=RSOS-180748>

- Competing interests

- Authors' contributions

- Acknowledgements

- Funding statement

Kind regards,

Andrew Dunn

on behalf of Prof. Jon Blundy (Subject Editor)

Associate Editor's comments:

Please accept our apologies for the unusual delay in completing this phase of the review process. Regrettably, we struggled to find sufficient referees to render a decision. Two reviewers have now reported. Please ensure that you fully address the concerns of the referees (in particular referee 2) during your revision, and be aware that further major revisions will be granted only in exceptional circumstances. We look forward to receiving your revision.

Comments to Author:

Reviewers' Comments to Author:

Reviewer: 1

Comments to the Author(s)

The manuscript is quite well written and almost no problems are found. But the size of letters used in Figures 1 and 2 is too small. The configurations of some molecules shown in Figure 1 are also too small.

Hidetoshi Okuyama, Formerly Hokkaido University.

Reviewer: 2

Comments to the Author(s)

The manuscript 'Public questions spur the discovery of new bacterial species able to degrade lignin in industrial waste' presents a new source of bacteria and, potentially enzymes, that may be involved in lignocellulose degradation. I think this work is interesting from an ecological and evolutive point of view. However, it is very basic from the microbial point of view and in the lignin depolymerization field. The work is well written but there are many statements that would need to be toned down, modified, or eliminated regarding the discovery of enzymes or the novelty of the work. The secretome results also need to be more critically discussed. During the last years, there has been a growing number of manuscripts on the field of bacterial lignin degradation that would need to be reviewed to present a more realistic discussion.

Specifics:

- The introduction is lacking information on what it is known regarding lignin depolymerization by bacteria. Please, include a paragraph about that. Below some key publications on this topic that contain a good source of references.

1. Huang XF et al, 2013. Isolation and characterization of lignin-degrading bacteria from rainforest soils. *Biotechnol Bioen*.
2. Linger et al, 2014. Lignin valorization through integrated biological funneling and chemical catalysis. *PNAS*.
3. Salvachua et al, 2015. Towards lignin consolidated bioprocessing: simultaneous lignin depolymerization and product generation by bacteria. *Green Chemistry*.
4. Lin et al. 2016 Systems biology-guided biodesign of consolidated lignin conversion. *Green Chemistry*.

- Page 5-line 7: what is exactly 'Biochoice lignin'? Is that a soluble or insoluble lignin stream?

- Page 5-line 6: Did the authors try higher black liquor concentrations with the bacteria selected?

- Considering that the paper is mostly focused in black liquor, a detailed composition of this substrate should be given. For instance, many black liquor streams contain carboxylic acids (i.e. acetate, glycolic acid, etc) and monomeric aromatic compounds (i.e. guaiacol, p-coumaric acid, ferulic acid, vanillic acid, etc) that can be utilized by bacteria as only carbon source. If this is the case, authors may be finding aromatic-catabolic bacteria and not lignin-degrading bacteria. Please present a table with the composition of the black liquor.

- Page 6, line 11. 'Turbid' is not a very precise way to present bacterial growth. If possible, present the values corresponding to optical density measurements (i.e. optical density at 600 nm).

- Page 6-line 45. It is very interesting (and not typical) to find bacteria that are able to grow at a such large range of pHs (4 to 10). Was the pH verified in the plates? It is worth considering that the buffering capacity of phosphate buffer (in M9) is between 6 and 8. Thus, different buffers might be preferred for pHs out of that range.

- Page 7-line 10. '... lignin as sole carbon source'. How was the lignin sterilized? Some sterilization processes (i.e autoclave) can release aromatic compounds from high molecular weight lignin and be utilized as carbon source by the bacteria. If this is the case, again, we cannot talk about lignin degradation only.

- Page 7-line 31- 'involved in the breakdown of black liquor'. To conclude this, the characterization of the black liquor (before and after the bacterial treatment) is needed. Again, were there aromatic compounds or other carbon sources apart from high molecular weight lignin? What was the lignin content? What was the size distribution of the lignin? If none of these parameters are given, sentences like this are recommended to be removed along the manuscript.

- Page 7, line 41, Table 4. There is a high number of cytoplasmic proteins in the secretome. Could that be just an effect of bacterial lysis and not secretion? Please comment on this.

- Page 8- line 27-line 31... It is highly recommended to further review the literature in this topic. There is a considerable number of bacteria able to utilize black liquors, kraft lignin, and the aromatic compounds on it. In fact, some of those bacteria are able to break down the lignin, which is not demonstrated in this work (see references above).

- Page 8, line 37. 'lignin as the sole carbon source'. Characterization is needed.

- Page 9-line 40. 'Brown rot fungi are known to hydrolyze lignin...'. Brown rot fungi do not break down lignin, they only modify it to attack the cellulose, that is why the residue after the fungal attack is brown (compared to white-rot fungi, which really degrade lignin). Please read book chapter from Barry Goodell entitled 'Brown-Rot Fungal Degradation of Wood: Our Evolving View'.

- Page 9-line 37. Authors present manganese superoxide dismutase as a potential enzyme involved in lignin depolymerization. Looking at Table 4, that enzyme was not even found in the lignin media. Discuss this critically. In addition, manganese superoxide dismutase is presented as a cytoplasmic protein. Lignin depolymerization is an extracellular process. Please comment on this as well.

- Table 4. Additional comments: there is another superoxide dismutase that appears in all media (also in lignin, 3835). This result suggests that this enzyme may not be involved in lignin break down and only be related with detoxification. Authors also include in Figure 1 a Glutathione-S-transferase (3813) as potential enzyme in the lignin degradation pathway. However, this enzyme was not found either in the secretomic study in the lignin media (only in cellulose media).

Typically, differential proteomic analyses give insights of the enzymes that are differentially expressed and needed to metabolize various substrates. However, these results are not conclusive and not critically discussed. Please consider all these comments to re-distribute the enzymes in the table (different sections) and be more critical in the discussion.

-Page 10-line 29- 'Here we have identified a novel source of bacterial enzymes involved in lignin degradation'. This statement is not correct until it is demonstrated (i.e. lignin characterization or enzymes assays).

-Page 10-line 35. Microbiologists are routinely looking into new sources of microorganisms and enzymes in different environments through bioprospecting. This is a very typical tool that this community uses and specifies in their papers. Saying that scientists 'focus on screening private culture collections' is not right. Please tone down. See references above and the one below.

-Page 10- line 40-41. The search of enzymes in insects or crustaceans has been reported years ago. Here just an example on a crustacean, the gribble, studied because of the habitat that colonized and in which powerful cellulases were discovered. Paper: King et al, 2010. Molecular insight into lignocellulose digestion by a marine isopod in the absence of gut microbes. PNAS.

-Page 10, line 1-4. Authors need to demonstrate lignin break down to make this statement.

Author's Response to Decision Letter for (RSOS-180748.R0)

See Appendix A.

RSOS-180748.R1 (Revision)

Review form: Reviewer 2

Is the manuscript scientifically sound in its present form?

Yes

Are the interpretations and conclusions justified by the results?

Yes

Is the language acceptable?

Yes

Is it clear how to access all supporting data?

Yes

Do you have any ethical concerns with this paper?

No

Have you any concerns about statistical analyses in this paper?

No

Recommendation?

Accept with minor revision (please list in comments)

Comments to the Author(s)

The manuscript has been greatly improved and most of the reviewer's comments addressed.

Minor comments:

- Reference 15 shows to the utilization of black liquor by bacteria, not Kraft lignin as indicated in Page 2, line 38. Please modify that.
- Page 9- line 18. "We also identified 9 new species able to break down alkaline black liquor (as demonstrated by growth on black liquor as the sole carbon source), which contains lignin and sugars." Bacteria may be utilizing low molecular weight lignin and sugars to support the growth. Thus, it may be convenient to replace "break down black liquor" by "utilize black liquor as carbon source, which contains lignin and sugars".

Decision letter (RSOS-180748.R1)

23-Jan-2019

Dear Dr Mathews:

On behalf of the Editors, I am pleased to inform you that your Manuscript RSOS-180748.R1 entitled "Public Questions Spur the Discovery of New Bacterial Species Associated with Lignin Bioconversion of Industrial Waste" has been accepted for publication in Royal Society Open Science subject to minor revision in accordance with the referee suggestions. Please find the referees' comments at the end of this email.

The reviewers and Subject Editor have recommended publication, but also suggest some minor revisions to your manuscript. Therefore, I invite you to respond to the comments and revise your manuscript.

- Ethics statement

- Data accessibility

If you wish to submit your supporting data or code to Dryad (<http://datadryad.org/>), or modify your current submission to dryad, please use the following link:
<http://datadryad.org/submit?journalID=RSOS&manu=RSOS-180748.R1>

- **Competing interests**

- **Authors' contributions**

- **Acknowledgements**

- **Funding statement**

Because the schedule for publication is very tight, it is a condition of publication that you submit the revised version of your manuscript before 01-Feb-2019. Please note that the revision deadline will expire at 00.00am on this date. If you do not think you will be able to meet this date please let me know immediately.

When submitting your revised manuscript, you will be able to respond to the comments made by the referees and upload a file "Response to Referees" in "Section 6 - File Upload". You can use this to document any changes you make to the original manuscript. In order to expedite the

processing of the revised manuscript, please be as specific as possible in your response to the referees.

on behalf of Professor Jon Blundy (Subject Editor)
openscience@royalsociety.org

Reviewer comments to Author:
Reviewer: 2

Comments to the Author(s)

The manuscript has been greatly improved and most of the reviewer's comments addressed.

Minor comments:

- Reference 15 shows to the utilization of black liquor by bacteria, not Kraft lignin as indicated in Page 2, line 38. Please modify that.
- Page 9- line 18. "We also identified 9 new species able to break down alkaline black liquor (as demonstrated by growth on black liquor as the sole carbon source), which contains lignin and sugars." Bacteria may be utilizing low molecular weight lignin and sugars to support the growth. Thus, it may be convenient to replace "break down black liquor" by "utilize black liquor as carbon source, which contains lignin and sugars".

Author's Response to Decision Letter for (RSOS-180748.R1)

See Appendix B.

Decision letter (RSOS-180748.R2)

07-Feb-2019

Dear Dr Mathews,

I am pleased to inform you that your manuscript entitled "Public Questions Spur the Discovery of New Bacterial Species Associated with Lignin Bioconversion of Industrial Waste" is now accepted for publication in Royal Society Open Science.

on behalf of Professor Jon Blundy (Subject Editor)
openscience@royalsociety.org

Appendix A

Editors comments and changes

1. The email for RK Blackburn has been updated and ScholarOne should not be in spam.

Please note that rkblackb@ncsu.edu appears to have identified ScholarOne emails as spam, and will not be able to receive this message. Please ensure this author is forwarded a copy of the decision and asked to resolve the blacklisting of ScholarOne (this may require interaction with ScholarOne and their institution).

2. Ethics statement is present on page 13 lines 2-3. Data accessibility statement is present on page 12 lines 45-46. Competing Interests is present in page 12 lines 32. Authors contributions is present on page 12 lines 34-40. Acknowledgements were not required. Funding Statement is present on page 12 lines 42-43 and was updated with funding sources for all authors.

- *Ethics statement (if applicable)*

- *Data accessibility*

It is a condition of publication that all supporting data are made available either as supplementary information or preferably in a suitable permanent repository. The data accessibility section should state where the article's supporting data can be accessed. This section should also include details, where possible of where to access other relevant research materials such as statistical tools, protocols, software etc can be accessed. If the data have been deposited in an external repository this section should list the database, accession number and link to the DOI for all data from the article that have been made publicly available. Data sets that have been deposited in an external repository and have a DOI should also be appropriately cited in the manuscript and included in

the reference list.

If you wish to submit your supporting data or code to Dryad (<http://datadryad.org/>), or modify your current submission to dryad, please use the following link: <http://datadryad.org/submit?journalID=RSOS&manu=RSOS-180748>

- **Competing interests**

- **Authors' contributions**

- **Acknowledgements**

- **Funding statement**

All changes made refer to those requested by reviewer 2 unless otherwise noted.

1. Text size in Figure 3 was increased as suggested by reviewer 1.
2. Many of the statements about degradation of lignin and black liquor were clarified.
 - a. The title was changed to better reflect the findings. We aim to follow this work with enzymatic characterization as an indicator of lignin degradation. Page 1 lines 1-2 clarify that we have found bacteria able to grow with lignin as a sole carbon source rather than indicating that the bacteria have been shown to degrade lignin.
 - b. Growth on lignin which suggests lignin degradation was highlighted in the abstract on page 1 line 30
 - c. Breakdown of black liquor was determined by growth as the sole carbon source and not confirmed with chemical characterization. Future research will use these techniques to identify enzymes from *C. lapagei* and the other arthropod isolates. Page 7 line 34
 - d. Biochoice lignin was autoclaved when added to water separately from the M9 salts solution. It is possible that during autoclaving lignin structure can be changed such that aromatics are released. We have now noted this in the discussion on page 9 lines 4-10. A reference for this information was also added. Page 15 lines 8-10
 - e. Lignin degradation, lignin tolerance, and aromatic utilization enzymes were identified in this work and this was clarified on page 10 lines 4-5.
 - f. Statements about the number of new bacteria that can degrade lignin and black liquor were further clarified since chemical characterization was lacking. Page 8 lines 33
 - g. "Here we have identified a novel source of bacterial enzymes involved in lignin degradation" was modified to read "... involved in lignin bioconversion". Page 11 lines 3-4.
3. The background was updated to include more recent information about lignin degradation as suggested. Page 2 lines 32-41 and references page 14 lines 5-21
4. Black liquor characterization was performed previously but details were added and paper referenced. Changes made on page 4 lines 42-43.

5. Laccase activity of *C. lapagei* was determined to better display lignin degradation ability.
 - a. Methods for this assay were added on page 6 lines 17-18.
 - b. Results of this assay are presented on page 8 lines 20-23 and in Figure 2.
 - c. The results were discussed on page 10 lines 5-9.
6. The descriptor "turbid" was removed from the description of inoculum used for secretome analysis. Overnight cultures were saturated and in stationary phase. The optical density of these cultures was not measured. Changes to reflect this information were made on page 6 lines 33-34.
7. A more detailed description of BioChoice Lignin and reference to the structural characterization was added on page 7 lines 31-34. The reference was added on page 16 lines 8-10.
8. Statements about the numbers of new species were toned down as the reviewer correctly pointed out that new species are being identified and published throughout the peer review process of this article. Page 9 lines 10-12.
9. Updated references were provided for new work on lignin degrading bacteria from termites. Page 9 line 17 and references page 16 lines 21-26.
10. Identification of proteins in the secretome is largely dependent on centrifugation after growth. It is possible that bacterial lysis could have occurred during growth such that this analysis also identified some proteins produced in the cytosol. Of the enzymes identified there were very few with secretion signals. It is also possible some of these proteins are secreted but secretion signals were not identified Page 10 lines 14-20 Supporting references can be found on page 17 lines 1-6.
11. Enzyme grouping in Table 4 were more critically discussed It is possible that many of the enzymes identified when grown on cellulose and hemicellulose are not directly involved in lignin degradation but instead detoxification. Page 10 lines 29-35.
12. Brown rot fungi have been shown to be involved in lignin depolymerization and not lignin degradation. This was further clarified page 10 lines 39-40.
13. Language about previous attempts to find bacteria or enzymes involved in this process was toned down and additional more recent efforts were included. Page 11, lines 37-40
14. New information was also added about lignocellulose degrading organisms that produce enzymes themselves rather than by relying

on their gut microbiota. Page 11 lines 43-45 and references page 18 lines 1-8

15. In text references were updated throughout the manuscript and well as in the tables.

Changes not made

1. Figure 1 was not modified as suggested by reviewer 1 to ensure a high resolution figure.
2. Higher amounts of black liquor were not tested due to pH of camel cricket gut. 1% black liquor was pH 11.5, 5% black liquor was pH 12 and 10% black liquor was pH 12.3 it did not seem likely that bacteria isolated from insects could survive at more basic pHs. Please note that previous work has isolated a bacterium directly from black liquor with an optimal pH of 9. Generation time of this bacterium at pH 12 was over 39 hours (Mathews et al. 2014).
3. The pH of the media for both Congo Red and M9 black liquor was measured to determine range of pH in which these bacterial isolates could grow. M9 was not the base for the Congo Red media and therefore buffering capacity is within range. pH measurements provided on page 5 lines 17-20.
4. Reviewer 2 commented that the statement on page 10 lines 29-32 about applied value of this information. However we believe that growth on cellulose, xylan, and biochoice lignin still demonstrate the value of *C. lapegei* in lignocellulose degradation and depolymerization for applications in the bioenergy, wood and pulping, textiles and chemical production industries.

Appendix B

Editors comments and changes

1. Acknowledgements were added

All changes made refer to those requested by reviewer 2 unless otherwise noted.

1. Reference 15 was modified to indicate that the subject was black liquor utilization and not kraft lignin. Page 2, line 38.
2. Page 9 line 13 was modified as suggested to read “utilize” alkaline black liquor as the carbon source which contains lignin and sugars.